# Vitamin B6, B12, and Folate’s Influence on Neural Networks in the UK Biobank Cohort

**DOI:** 10.3390/nu16132050

**Published:** 2024-06-27

**Authors:** Tianqi Li, Juan Pedro Steibel, Auriel A. Willette

**Affiliations:** 1Genetics and Genomics Program, Iowa State University, Ames, IA 50011, USA; tianqili@iastate.edu; 2Department of Animal Science, Iowa State University, Ames, IA 50011, USA; jsteibel@iastate.edu; 3Department of Neurology, Rutgers University, New Brunswick, NJ 07101, USA

**Keywords:** normal aging, functional connectivity, diet

## Abstract

Background: One-carbon metabolism coenzymes may influence brain aging in cognitively unimpaired adults. Methods: Baseline data were used from the UK Biobank cohort. Estimated intake of vitamin B6, B12, and folate was regressed onto neural network functional connectivity in five resting-state neural networks. Linear mixed models tested coenzyme main effects and interactions with Alzheimer’s disease (AD) risk factors. Results: Increased B6 and B12 estimated intake were linked with less functional connectivity in most networks, including the posterior portion of the Default Mode Network. Conversely, higher folate was related to more connectivity in similar networks. AD family history modulated these associations: Increased estimated intake was positively associated with stronger connectivity in the Primary Visual Network and Posterior Default Mode Network in participants with an AD family history. In contrast, increased vitamin B12 estimated intake was associated with less connectivity in the Primary Visual Network and the Cerebello–Thalamo–Cortical Network in those without an AD family history. Conclusions: The differential patterns of association between B vitamins and resting-state brain activity may be important in understanding AD-related changes in the brain. Notably, AD family history appears to play a key role in modulating these relationships.

## 1. Introduction

All-cause dementia is a broad category of neurodegenerative disorders, characterized by progressive deterioration in cognition and neuronal loss in highly stereotyped areas. Globally, dementia cases are projected to surge from 57.4 million in 2019 to an estimated 152.8 million by 2050 [1]. Alzheimer’s disease (AD), accounting for 60–80% of dementia diagnoses, is marked by progressive memory loss, deficits in other cognitive domains, deposition of beta-amyloid (Aβ) and hyperphosphorylated tau, and behavioral disturbances. AD-related brain atrophy typically begins in the entorhinal cortex and hippocampus, with subsequent gray matter loss in the frontal and parietal regions depending on the AD subtype [2]. These regions are crucial as the entorhinal cortex and hippocampus are key to memory processing. Their degeneration directly correlates with hallmark memory impairments seen in AD patients.

Diet is an important modifiable risk factor for all-cause dementia and particularly AD [3,4]. Previous studies have shown that consumption of certain foods such as red wine, milk, and cheese is related to fluctuations in resting-state functional connectivity in brain networks involved in executive function and the Posterior Default Mode Network [5]. Notably, some connectivity patterns associated with cheese consumption were influenced by AD family history and genetic factors like the Apolipoprotein E ε4 (APOE4) genotype, which is the strongest genetic risk factor for AD [6]. Additionally, variations in the Translocase of Outer Mitochondrial Membrane-40 (TOMM40) gene, adjacent to APOE on Chromosome 19, may also be implicated in AD [7]. The current study aims to extend these findings by examining how different nutritional compounds, specifically B vitamins, could serve as biomarkers to track resting-state functional connectivity patterns.

Vitamins are vital organic compounds that are crucial for normal human physiology. In particular, metabolic pathways essential for cellular functions [8]. Studies indicates that pyridoxine hydrochloride (“vitamin B6”), hydroxycobalamin (“vitamin B12”), and folate may have a role in neurodegenerative processes. Briefly, these B vitamins are important for modifying the demethylation of methionine to homocysteine, a sulfur-containing amino acid that is a reliable biomarker of cardio- and cerebrovascular events like heart attacks and strokes [9]. It is thought that vitamin B supplementation reduces homocysteine levels and could potentially attenuate cognitive decline seen in cognitively unimpaired older adults. For cognition, homocysteine but not B vitamin levels has been associated with cross-sectional [10] or longitudinal deficits [11] in global cognition or cognitive domains like executive function and declarative memory. More recent meta-analyses suggest an overall modest improvement in global cognition but lower all-cause dementia risk with higher folate intake [12]. However, randomized clinical trials offer mixed results for the benefits of vitamin B6, B12, and folic acid supplementation [8,13].

Resting-state functional magnetic resonance imaging (fMRI) can predict cognitive or affect behaviors [14] and detect preliminary changes in brain function that may signify cognitive impairment related to AD [15]. Studies employing independent component analysis have identified variation in connectivity patterns among individuals diagnosed with Mild Cognitive Impairment (MCI) or AD [16]. Our previous work has drawn connections between AD risk factors, cognitive impairment, and fMRI results by evaluating resting-state functional connectivity [17,18].

Additionally, task-based fMRI studies have provided insights into how brain activity, influenced by vitamin levels, predicts cognitive functions. A recent study found that the hemodynamic activity of the right dorsal anterior cingulate cortex was positively associated with vitamin B12 concentrations [19]. This region’s activity significantly predicted subjects’ visual search and attention abilities, highlighting the role of brain imaging as an intermediate phenotype that links vitamin concentrations to cognitive profiles in elders. These findings underscore the importance of considering brain activity in addition to blood biochemistry when evaluating the impact of vitamins on cognitive health.

To enhance our understanding of the influence of B vitamins on cognitive health, we explored data from a subset of 12,025 UK Biobank participants at baseline. Our focus was on scrutinizing the relationship between intake of vitamins B6, B12, and folate on degree of neural network functional connectivity related to visual processing and cognition. This examination was contextualized by further considering a spectrum of AD risk factors, including APOE4 and TOMM40 genotypes, as well as AD family history.

## 2. Materials and Methods

### 2.1. Cohort and Participants

Baseline UK Biobank data comprised approximately 500,000 individuals ranging in age from 40 to 70 years. These participants were recruited from 22 evaluation facilities across the United Kingdom [20]. We utilized data from 12,025 participants without neurological disorders (excluding conditions such as central nervous system diseases, cerebrovascular diseases, and mental and behavioral disorders) [5]. These individuals had data available on genomics, resting-state fMRI, vitamin B intake, and demographics.

### 2.2. Resting-State fMRI

Participants were scanned at one of three designated sites: Reading, Newcastle, or Manchester. These scans were conducted using a Siemens Skyra 3T system, fitted with a 32-channel RF receiver head coil, provided by Siemens Medical Solutions based in Erlangen, Germany [21]. Initial MRI evaluations started in 2014, with an ongoing process of longitudinal data collection [22]. The scanning procedure has been described [5,17,18]. For insights into preprocessing and quality control measures, please consult: https://biobank.ctsu.ox.ac.uk/crystal/crystal/docs/brain_mri.pdf (accessed on 28 May 2024).

FMRIB’s MELODIC was used for both group Principal Component Analysis and Independent Component Analysis. A total of 21 spatially orthogonal, non-noise, distinct Independent Components (ICs) were identified to represent resting neural networks [23]. The Papaya viewer allows for online visualization of these ICs: https://www.fmrib.ox.ac.uk/ukbiobank/group_means/rfMRI_ICA_d25_good_nodes.html (accessed on 28 May 2024).

Each participant’s EPI scan underwent a spatial back projection of an IC to determine intrinsic functional connectivity. The primary T-value map was used to gauge mean activation level. In turn, signal was then transformed into a Z-score for a more straightforward interpretation. As described, an expert (AAW) analyzed the activation maps and identified underlying neural networks [5].

This study primarily centered on five networks related to cognitive and visual processing functions, as detailed in Appendix A.

### 2.3. Genetic Factors—APOE, TOMM40, and AD Family History

Isoforms of the APOE haplotype (ε2, ε3, and ε4) were determined through the SNPs rs429358 and rs7412. Based on the presence or absence of at least one ε4 allele, participants were classified into two categories: ε4 carriers (with configurations ε2/ε4, ε3/ε4, ε4/ε4) or ε4 non-carriers (with configurations ε2/ε2, ε2/ε3, ε3/ε3).

As described [18], we extracted TOMM40 genotype data specific to the rs2075650 (SNP ‘650) using PLINK version 1.90, accessible at https://www.cog-genomics.org/plink/1.9/ (accessed on 28 May 2024). For TOMM40 ‘650 status, individuals were categorized as non-G carriers (i.e., those with AA homozygosity) or G-carriers (i.e., encompassing either GA or GG configurations).

The classification of AD family history was based on participants’ self-reports regarding AD family lineage from a touchscreen questionnaire. The query was: “Has/did your father/mother ever suffer from:”, succeeded by a roster of chronic ailments. Among these options was ‘Alzheimer’s disease/dementia’.

### 2.4. Covariates

Covariates included a participant’s baseline age (in years) and gender (either male or female). Additional covariates included alcohol consumption (never, past, or current drinker), smoking (never, former, or active) [18], Body Mass Index (BMI), and the Townsend Index to gauge socio-economic classification [5]. This index was derived from national census output regions and established for each participant prior to their involvement in the UK Biobank. Scores assigned to participants mirrored those of their respective postcode’s output zones.

### 2.5. Vitamin B Intake

The estimated intake of vitamin B6, vitamin B12, and folate was based on participants’ responses to a dietary questionnaire. Specifically, participants filled out a comprehensive 24 h dietary questionnaire, detailing up to 206 different food items and 32 beverages consumed in the last 24 h [24,25]. The questionnaire can be reviewed at https://biobank.ctsu.ox.ac.uk/crystal/crystal/docs/DietWebQ.pdf (as of 28 May 2024). The computation of nutrient intake used the Oxford WebQ data from the UK’s McCance and Widdowson’s food composition table (FCT) and its accompanying supplementary data [26]. Participants who took vitamin B supplements were excluded from the analysis.

### 2.6. Statistical Analyses

Data processing and analytics were performed using R, version 4.3.2 (RStudio, Posit Software, Boston, MA, USA). Linear mixed models examined how dietary habits were related to each resting-state IC. Mixed model analyses were conducted using the lmerTest R package.

In this analytical framework, the location of the UK Biobank assessment center, where participants provided their consent, is employed as a random effects element in the model. This approach addresses variability that may stem not from the participants’ inherent differences but from the specific environments of the assessment centers. Outcomes included the 21 resting-state ICs. Independent variables were estimated intake of vitamin B6, B12, or folate consumed by the participants based on a dietary questionnaire. Other fixed effects included APOE and TOMM40 genotypes, family history of AD, sex, age, Townsend index, smoking, and alcohol status. Exploratory analyses examined interactions between APOE, TOMM40, and family history of AD on outcomes of interest. This was to evaluate if genetic variables moderated associations between estimated vitamin intake and neural network functional connectivity. Alpha was set at 0.05. Due to potential decreases in statistical Power for interaction effects, a more accommodating alpha level of 0.10 was adopted [27,28]. To curtail the likelihood of type 1 error, a Benjamini–Hochberg procedure was initially conducted [29].

## 3. Results

### 3.1. Demographics and Data Summaries

Table 1 summarizes demographic and adjunctive information. Appendix A describes our five neural networks of interest.

### 3.2. Main Effects

Higher estimated intake of vitamin B6 was related to less functional connectivity in several neural networks: the Dorsal and Ventral Stream Visual Network (IC 2, beta = −0.0115 *p* = 0.017), the Visuocerebellar Network (IC 4, beta = −0.0125, *p* = 0.006), the Primary Visual Network (IC 8, beta = −0.023, *p* < 0.001), the Cerebello–Thalamo–Cortical Network (IC 19, beta = −0.023 *p* < 0.001), and the Posterior Default Mode Network (IC 20, beta = −0.011 *p* < 0.001).

Similarly, higher intake of vitamin B12 was linked with less connectivity in the following networks: the Dorsal and Ventral Stream Visual Network (IC 2, beta value −0.002, *p* = 0.005) (Figure 1), the Visuocerebellar Network (beta value −0.002, IC 4, *p* = 0.01), the Primary Visual Network (IC 8, beta value −0.003, *p* = 0.007), the Cerebello–Thalamo–Cortical Network (IC 19, beta value −0.003, *p* < 0.001), and the Posterior Default Mode Network (IC 20, beta value −0.002, *p* < 0.001).

On the other hand, greater folate consumption was correlated with more connectivity in several networks: the Visuocerebellar Network (IC 4, beta value = 0.0002, *p* = 0.014), the Primary Visual Network (IC 8, beta value = 0.0002, *p* = 0.013), the Cerebello–Thalamo–Cortical Network (IC 19, beta value = 0.0002, *p* = 0.007) (Figure 2), and the Posterior Default Mode Network (IC 20, beta value = 0.0002, *p* < 0.001).

### 3.3. Vitamin B Intake by Family History Interactions

Increased folate intake was positively associated with functional connectivity in the Primary Visual Network (IC 8) and the Posterior Default Mode Network (IC 20), regardless of family history of AD (see Table 2). However, the strength of this association varied between participants with and without a family history of AD. Participants with a family history of AD (i.e., FH+) showed a trend where higher folate intake correlated with increased neural network activity, as depicted in Figure 3. Specifically, the estimated beta value for the Primary Visual Network (IC 8) was 4.76-fold greater in FH+ participants than in those without a family history of AD (FH-) (beta estimates 0.000514 vs. 0.000108). Similarly, the estimated beta value for the Posterior Default Mode Network (IC 20) was 9.09-fold greater in FH+ participants compared to FH- participants (beta estimates 0.000291 vs. 0.000032).

For vitamin B12 intake, its effects differed based on AD family history. For participants without a family history of AD (i.e., FH-), increased vitamin B12 intake was associated with lower functional connectivity in the Primary Visual Network (IC 8) (Beta value = −0.0051, *p* < 0.001) and the Cerebello–Thalamo–Cortical Network (IC 19) (beta value = −0.0050, *p* < 0.001), as illustrated in Figure 4 and Figure 5, respectively. Conversely, for participants with a family history of AD (FH+), there was a positive trend between vitamin B12 intake and connectivity in IC 8 (beta value = 0.0007; *p* = 0.4703), and a negative trend for IC 19 (beta value −0.0006; *p* = 0.4549), although these trends were not statistically significant. In other words, the correlation between vitamin B12 intake and functional connectivity in the IC 8 and IC 19 networks did not reach statistical significance.

## 4. Discussion

The objective of this research was to explore the relationship between vitamin B (B6, B12, and folate) intake and functional connectivity in neural networks implicated in cognitive and visual processes, while accounting for influences from APOE4, TOMM40 genotypes, and family history of AD.

Our core findings indicate that higher intake of vitamin B6 and B12 was associated with decreased connectivity in several neural networks: the Dorsal and Ventral Stream Visual Network, the Visuocerebellar Network, Primary Visual Network, the Cerebello–Thalamo–Cortical Network, and the Posterior Default Mode Network. Conversely, an increased intake of folate was found to be positively correlated with connectivity in the Visuocerebellar Network, the Primary Visual Network, the Cerebello–Thalamo–Cortical Network, and the Posterior Default Mode Network.

Vitamin B6, B12, and folate are involved in the one-carbon metabolic pathway and are implicated in managing homocysteine levels, which, when elevated, can lead to endothelial dysfunction and increased brain lesion load (i.e., white matter hyperintensities) [30,31]. Findings are mixed for whether B vitamin supplementation can enhance cognitive function and prevent dementia [8,12,32,33]. Some research highlights potential benefits in reducing or delaying cognitive decline, particularly with folate supplementation [32]. However, clinical trials are inconclusive on whether B-vitamin supplementation lowers homocysteine levels and improves cognitive outcomes [34,35].

Vitamin B6. For vitamin B6, our results showed that higher intake was related to reduced connectivity in the all five examined networks: the Dorsal and Ventral Stream Visual Network (IC 2), the Visuo-cerebellar Network (IC 4), the Primary Visual Network (IC 8), the Cerebello–Thalamo–Cortical Network (IC 19), and the Posterior Default Mode Network (IC 20). These findings align with some clinical reports. Several case reports indicated sensory neuropathy symptoms in individuals consuming excessive vitamin B6 over extended periods, characterized by impairments in sensory functions like touch, temperature, and proprioception [36]. Small-fiber dysfunction may be an early clinical sign, prompting the European Food Safety Authority to set a Tolerable Upper Intake Level of 12 mg/day for vitamin B6 [37,38].

Vitamin B12. For vitamin B12, supplementation may have limited cognitive benefits [39,40,41,42,43,44]. Our findings indicate that increased vitamin B12 intake mirrored the outcomes of vitamin B6, with reduced connectivity in the Dorsal and Ventral Stream Visual Network (IC 2), the Visuocerebellar Network (IC 4), the Primary Visual Network (IC 8), the Cerebello–Thalamo–Cortical Network (IC 19), and the Posterior Default Mode Network (IC 20). However, the impact of vitamin B12 was significantly influenced by AD family history, but not APOE4 or TOMM40 genotypes. Specifically, only participants with a family history of AD who consumed more vitamin B12 exhibited more neural network connectivity in visual networks. This suggests that the cognitive effects of vitamin B12 may depend on genetic factors beyond major age-related genes on Chromosome 19 (i.e., APOE4 and TOMM40).

Folate. For folate, our findings indicate that, in terms of main effects, increased folate intake is associated with more connectivity in several neural networks: the Visuocerebellar Network, the Primary Visual Network, the Cerebello–Thalamo–Cortical Network, and the Posterior Default Mode Network. Regarding interaction effects, family history appears to play a notable role in the impact of folate intake. Specifically, participants with AD family history have a much stronger relationship between more folate intake and more functional connectivity in the Primary Visual Network and the Posterior Default Mode Network.

The role of folate in dementia remains a topic of debate. Folate acts as a coenzyme or cosubstrate in single-carbon transfers that are critical in the synthesis of nucleic acids and amino acid metabolism. One of the key folate-dependent reactions is the transformation of homocysteine into methionine, which plays a part in synthesizing S-adenosyl-methionine, a significant methyl donor [45]. High levels of homocysteine have a consistent association with cardio- and cerebrovascular occlusions and disease [46,47]. Folate and B12 are closely linked in these methylation reactions. Given this interplay, there is a concern that high doses of folic acid could obscure the harmful effects of vitamin B12 deficiency. Consequently, the National Institutes of Health (NIH) advise that folic acid intake from fortified food and supplements should not exceed 1000 μg daily in healthy adults [48].

However, several studies have identified a positive relationship between high homocysteine levels and the incidence of Alzheimer’s disease-related dementia. Additionally, some observational studies have found links between low serum folate concentrations and both diminished cognitive function and an increased risk of Alzheimer’s disease-related dementia [32,49,50,51]. Nevertheless, even though folic acid supplementation lowers homocysteine levels, most clinical trials have not shown that folic acid supplementation significantly impacts cognitive function or the development of Alzheimer’s disease [52].

Our study has several limitations that warrant acknowledgment. Firstly, the data on vitamin B intake were derived from self-reported dietary information obtained from the UK Biobank, which could introduce recall bias and potential inaccuracies. To address the skewed distribution of vitamin B intake data, various transformations were considered to provide a more accurate representation of the data distribution and mitigate potential biases in our analysis. We applied both log-transformation and boxcox-tranformation, but neither completely resolved the skewness. However, the analyses using both transformed datasets showed similar patterns of association, indicating that the skewness did not significantly affect our interpretations (Appendix A).

Despite controlling for numerous potential confounding variables, it is possible that other unmeasured factors, such as physical activity levels and overall diet quality, might influence the observed associations. Additionally, the demographic composition of the UK Biobank participants may not fully represent the diversity of the broader population, which could limit the generalizability of our findings.

Furthermore, while our study focuses on the relationship between vitamin B intake, genetic factors, and resting-state functional connectivity, additional analyses could provide deeper insights. Future research should consider conducting subgroup analyses based on demographic factors such as age groups, gender, and socio-economic status to identify if certain subgroups are more affected by vitamin B intake and genetic factors in relation to neural connectivity. Additionally, while this study did not include the impact of detailed dietary patterns, such as meat consumption or specific dietary practices (e.g., vegan, vegetarian), future research should examine these factors.

Supplement use was not deeply analyzed in this study due to the binary nature of the available data. Although our current study is cross-sectional, incorporating longitudinal variables in future research could provide insights into how dietary and genetic factors influence the progression of neural connectivity changes over time. This approach could help to understand the dynamics of cognitive decline better and establish more precise biomarkers and intervention strategies for preventing or mitigating cognitive impairment related to AD.

Finally, while our study primarily relied on self-reported dietary intake, future research should incorporate biomarkers for vitamins to validate and complement self-reported data. This would enhance the accuracy of dietary assessments and provide a more robust basis for understanding the relationship between vitamin intake and neural connectivity.

## 5. Conclusions

Future research should strive to include more ethnically diverse populations to enhance understanding of the associations under study. There may be different relationships between vitamin B intake, AD risk factors, and cognitive performance. Such studies would contribute to the formulation of evidence-based dietary guidelines and interventions aimed at maintaining cognitive function and mitigating the risk of AD and other age-related cognitive impairments. In addition, as family history but not APOE or TOMM40 genotypes influenced B vitamin associations, it will be worthwhile to more broadly test for genetic factors that might influence vitamin B absorption and use in cognitive health.

## Figures and Tables

**Figure 1 nutrients-16-02050-f001:**
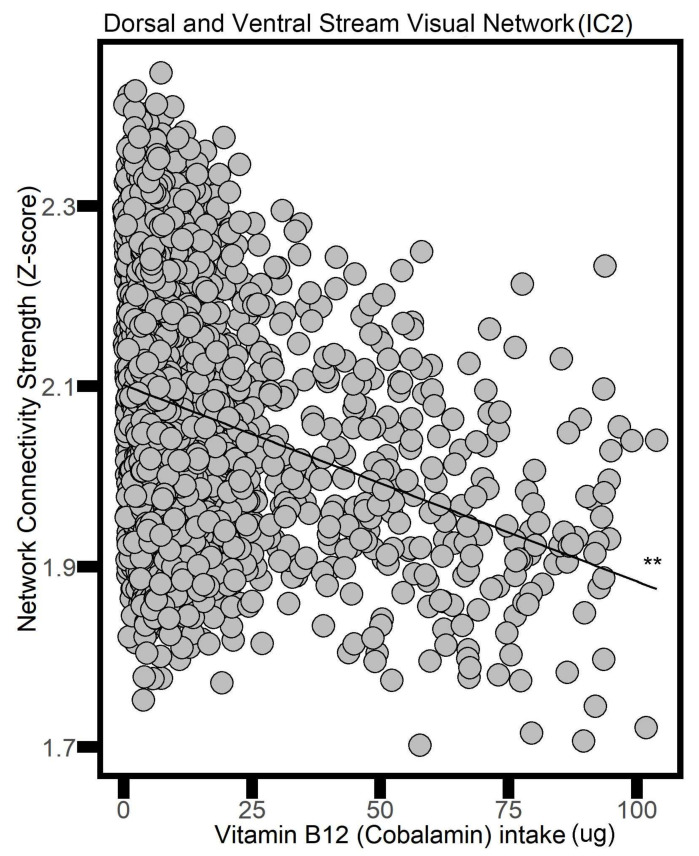
The association between vitamin B12 intake and the Dorsal and Ventral Stream Visual Network (i.e., neural network activity) in adults. ** *p* < 0.01.

**Figure 2 nutrients-16-02050-f002:**
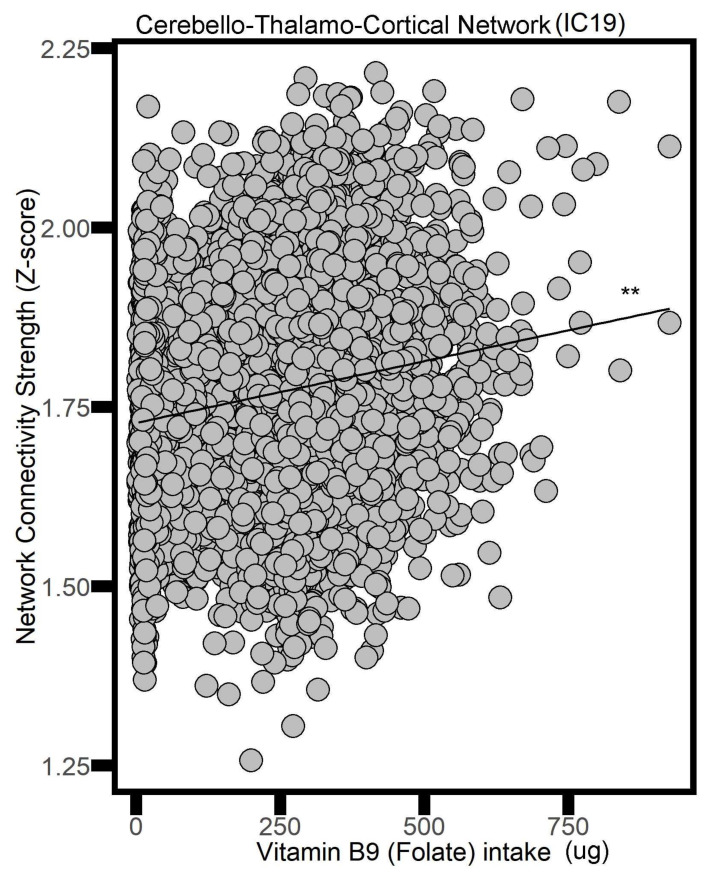
The association between folate intake and the Cerebello–Thalamo–Cortical Network (i.e., neural network activity) in adults. ** *p* < 0.01.

**Figure 3 nutrients-16-02050-f003:**
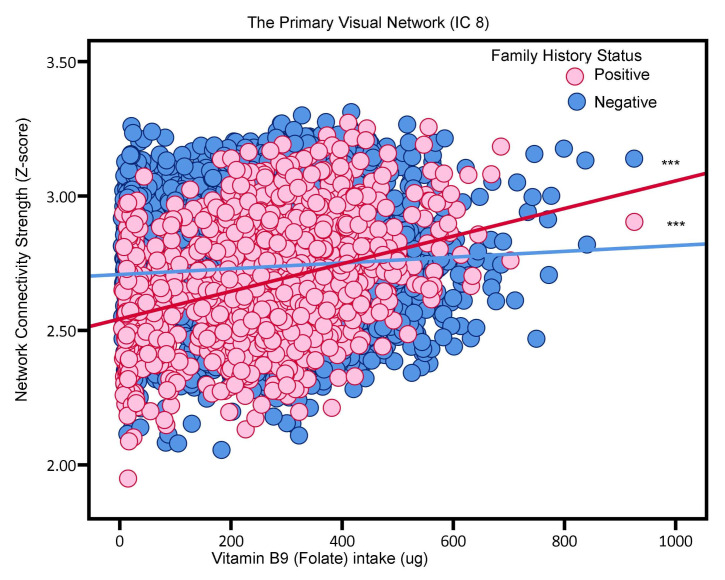
The association between folate intake and the Primary Visual Network (i.e., neural network activity) in adults without or with AD Family History (“positive”, “negative”). Blue and red, respectively, represent Family History negative and Family History positive participants. *** *p* < 0.001.

**Figure 4 nutrients-16-02050-f004:**
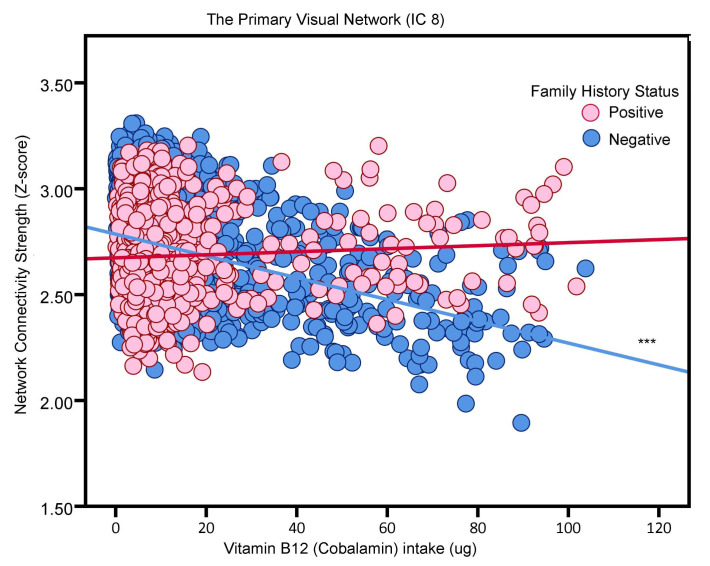
The association between vitamin B12 and the Primary Visual Network (i.e., neural network activity) in adults without or with AD family history (“positive”, “negative”). Blue and red, respectively, represent family history negative and family history positive participants. *** *p* < 0.001.

**Figure 5 nutrients-16-02050-f005:**
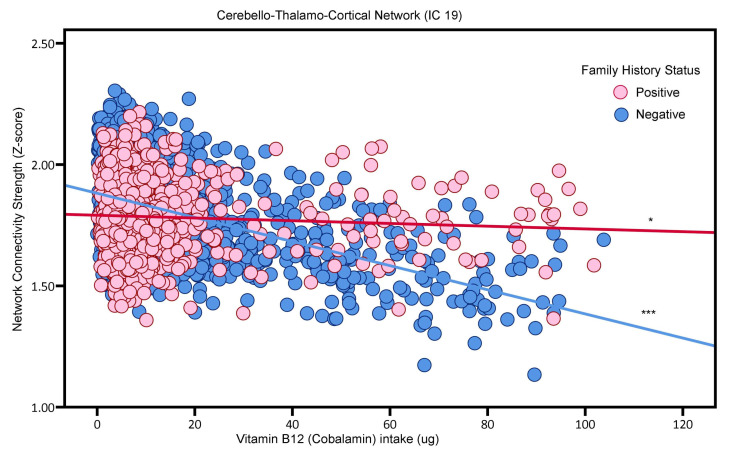
The association between vitamin B12 and the Posterior Default Mode Network (i.e., neural network activity) in adults without or with AD Family History (“positive”, “negative”). Blue and red, respectively, represent Family History negative and Family History positive participants. * *p* < 0.05 *** *p* < 0.001.

**Table 1 nutrients-16-02050-t001:** Demographic Characteristics of Participants.

Characteristic		
Baseline Age, mean (SD), y	55.07 (7.48)	Range: 40–70
Body Mass Index (BMI), mean (SD), kg/m^2^	26.59 (4.17)	Range: 14.74–56.12
Female, %	52.54	
APOE ε4 Status, %	27.68	
TOMM40 ‘650 Status, %	26.57	
Family History of AD, %	24.26	
Smoking Status, %		
Never	60.74	
Previous	32.89	
Current	6.37	
Alcohol Status, %		
Never	2.45	
Previous	1.95	
Current	95.60	
Vitamin B6, mean (SD), mg	2.63 (1.93)	Range: 0.02–19.87
Vitamin B12, mean (SD), ug	9.52 (12.74)	Range: 0.03–107.84
Folate, mean (SD), ug	259.80 (136.62)	Range: 3.65–1190.12

AD = Alzheimer’s disease; APOE = Apolipoprotein E. TOMM40 = Translocase of Outer Mitochondrial Membrane 40. All measures were obtained at baseline, with the exception of vitamin intake (the mean over five visits).

**Table 2 nutrients-16-02050-t002:** Estimates for folate and vitamin B12 intake interactions by risk factors.

Component	Folate				Vitamin B12			
	Family History Negative		Family History Positive		Family History Negative		Family History Positive	
	**Beta**	***p*-Value**	**Beta**	***p*-Value**	**Beta**	***p*-Value**	**Beta**	***p*-Value**
IC2	0.0001	0.0653	0.0002	0.0038	−0.0031	0.0000	−0.0002	0.7609
IC4	0.0001	0.1086	0.0002	0.0098	−0.0030	0.0000	0.0003	0.6989
IC8	**0.0001 ***	**0.0665**	**0.0005 ***	**0.0000**	**−0.0051 ***	**0.0000**	**0.0007 ***	**0.4703**
IC19	0.0003	0.0000	0.0002	0.0437	**−0.0050 ***	**0.0000**	**−0.0006 ***	**0.4549**
IC20	**0.0000 ****	**0.2299**	**0.0003 ****	**0.0000**	−0.0023	0.0000	0.0007	0.1609

Bolded text denotes *p* < 0.05. Neural networks with significant interactions; * *p* < 0.05, ** *p* < 0.01.

## Data Availability

The data in this study are owned by the UK Biobank (www.ukbiobank.ac.uk (accessed on 28 May 2024)), a data repository that can be accessed by applying through the UK Biobank Access Management System (www.ukbiobank.ac.uk/register-apply (accessed on 28 May 2024)). Due to the legal agreement, as researchers, we do not have permission to share the data, and we are not entitled to republish or otherwise make available any UK Biobank data at the individual participant level. All analyses and intellectual content separate from UK Biobank are available on a request basis.

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
