# Peer review of "Vitamin B6, B12, and Folate’s Influence on Neural Networks in the UK Biobank Cohort"

_nutrients, 2024, doi:10.3390/nu16132050_

Round 1

Reviewer 1 Report

Comments and Suggestions for Authors

Li and colleagues investigated the relationship between the levels of vitamins B6, B12, and folate and neural network functional connectivity in the context of Alzheimer's disease and cognitive health. Using data from 12,025 participants in the UK Biobank, the study aimed to elucidate how these vitamins influence brain connectivity patterns associated with visual processing and cognition. The research contextualizes these findings by considering various AD risk factors, including APOE4 and TOMM40 genotypes, as well as family history of AD. I only have a few comments to enhance the impact of the study.

1. The introduction provides a comprehensive overview of the relevance of diet and vitamins in AD and dementia. However, the section could benefit from a more structured flow, clearer definitions, and a stronger connection between the different components discussed. The discussion of previous findings related to diet and functional connectivity (lines 36-40) is important but feels somewhat isolated. The authors miss to discuss recent findings with task-based fMRI that are instead very relevant to their study and should be discussed (Cecchetti, L., Lettieri, G., Handjaras, G., Leo, A., Ricciardi, E., Pietrini, P., ... & Train the Brain Consortium. (2019). Brain Hemodynamic Intermediate Phenotype Links Vitamin B 12 to Cognitive Profile of Healthy and Mild Cognitive Impaired Subjects. Neural Plasticity, 2019.). Also, when mentioning specific brain regions (lines 32-34), authors should ensure that readers unfamiliar with neuroanatomy understand their significance. A brief explanation of why these regions are important in AD would be beneficial. The same goes for the discussion of genetic factors such as APOE4 and TOMM40 (lines 41-44), which could benefit from a bit more context on why these genes are important in AD research.

2. The current analysis is robust, focusing on the relationship between vitamin B intake, genetic factors, and resting-state functional connectivity. However, additional analyses could provide deeper insights and strengthen the findings. For instance, subgroup analyses based on different demographic factors such as age groups, gender, or socio-economic status could help identify if certain subgroups are more affected by vitamin B intake and genetic factors in relation to neural connectivity. Also, if longitudinal data are available, the authors should consider examining changes in functional connectivity over time in relation to vitamin B intake and genetic factors. This could provide insights into how dietary and genetic factors influence the progression of neural connectivity changes.

3. The Discussion section is well-organized but could benefit from clearer transitions between sections to improve readability. The authors should consider using subheadings for each vitamin to structure the discussion more effectively. The potential for recall bias in dietary data is noted, but the authors could suggest ways future studies could mitigate this, such as using biomarkers for vitamin levels or more frequent dietary assessments.

Comments on the Quality of English Language

The quality of English is good

Author Response

We are thankful to the reviewers for their detailed comments and suggestions. The manuscript has been much improved as a result.

1. Response: We appreciate the reviewer’s detailed feedback for the introduction. To address these comments, we have restructured the introduction to enhance the flow and clarity of definitions. We have also included a discussion of the recent findings by Cecchetti [1] to provide a more comprehensive context. Additionally, we have added brief explanations for the significance of the brain regions mentioned and elaborated on the importance of genetic factors such as APOE4 and TOMM40 in AD research.

Changes Made:

  1. Reorganized the introduction to improve flow and connectivity between topics.

  2. Included a discussion of Cecchetti, L. et. al. [1]to relate task-based fMRI findings to our study.

  3. Provided brief explanations of the significance of specific brain regions in AD.

  4. Expanded the discussion on genetic factors (APOE4 and TOMM40) to contextualize their relevance in AD research.

2. Response: We appreciate the valuable suggestions to enhance the analysis. Age, gender, and socio-economic status are included as covariates in our current study model. We avoided using these measures as independent variables in additional analyses, because it might confuse a general readership. Our primary focus was on genetic variables such as APOE and TOMM40 genotypes and family history of AD, building on our recent previous research [2]. Nonetheless, we agree that incorporating additional variables, such as demographic factors, can enrich the study and provide deeper insights. We have included this as a suggestion in the future directions portion of the Discussion section. Additionally, we acknowledge the potential benefits of subgroup analyses based on demographic factors and will consider these in future work. Our other study, which focuses on the biomarker IGF-1, has previously discussed the roles of variables like sex and age [3].

Finally, a longitudinal analysis is the next logical step to take with the resting state functional connectivity outcomes. For UK Biobank, there is unfortunately a precipitous drop-off in sample size for neuroimaging variables from baseline to subsequent waves. This is because the baseline study had 22 open sites for recruitment and processing, whereas now there is only 1 open site. Thus, our current dataset does not include longitudinal measures and discuss this limitation in the Discussion section. In the future research section, we suggest methodologies for incorporating longitudinal data.

Changes Made:

  1. Expanded the limitations and future research sections to include the value of subgroup analyses based on demographic factors.

  2. Suggested methodologies for future studies to examine changes in functional connectivity over time with longitudinal variables.

3. Response: We appreciate the feedback on improving the readability of the Discussion section. To address this, we have added subheadings for each vitamin to provide a clearer structure. We have also expanded the discussion on potential recall bias and suggested ways future studies could mitigate this issue, including the use of biomarkers and more frequent dietary assessments.

Changes Made:

  1. Added subheadings for each vitamin in the Discussion section.
  2. Expanded on the potential for recall bias and suggested methods to mitigate it in future research.

reference

  1. Cecchetti, L., et al., Brain Hemodynamic Intermediate Phenotype Links Vitamin B(12) to Cognitive Profile of Healthy and Mild Cognitive Impaired Subjects. Neural Plast, 2019. 2019: p. 6874805.
  2. Li, T., et al., Alzheimer's Disease Genetic Influences Impact the Associations between Diet and Resting-State Functional Connectivity: A Study from the UK Biobank. Nutrients, 2023. 15(15).
  3. Li, T., et al., Associations Between Insulin-Like Growth Factor-1 and Resting-State Functional Connectivity in Cognitively Unimpaired Midlife Adults. J Alzheimers Dis, 2023.

Reviewer 2 Report

Comments and Suggestions for Authors

The main concern of this reviewer is the treatment of dietary data - it should be done differently and it is therefore difficult to assess the quality of the rest of the manuscript:

- the Biobank dietary assessment has some limitations and authors should explore whether there are any studies that show reliability of B-vitamin intake assessment

- authors must be clear in their language and refer to "intake" and not "level" - this reviewer would recommend using "estimated intake"

- please consider log-transformation of intake data. The scatterplots suggest a log-normal distribution 

- please consider including other dietary factors, i.e. could B12 intake simply be a marker of meat intake, are there specific dietary patterns that might affect the association (e.g. vegan, vegetarian ...)

- did authors consider supplement use, which would be in particular relevant

- are any biomarkers for vitamins available that could be used?

Comments on the Quality of English Language

N/a

Author Response

Response: We thank the reviewer for their thorough and insightful feedback regarding the treatment of dietary data. We appreciate their comments and have addressed each point, as follows:

  1. Biobank Dietary Assessment Reliability:

We agree that the reliability of the Biobank dietary assessment should be critically evaluated. Several studies have used different approaches to discuss the association between dietary patterns and health conditions using the UK Biobank dataset (Brayner et al., 2021; Che et al., 2022; Kelly et al., 2023; Ling et al., 2024; Wang et al., 2023; Zhang et al., 2023). Some of these studies employed the same nutritional calculations from the questionnaire (Brayner et al., 2021; Kelly et al., 2023; Perez-Cornago et al., 2021; Zhang et al., 2023). Our study focuses on the association between daily dietary patterns and neural network activity. Although other nutritional assessments like genetically predicted plasma concentrations exist (Wang et al., 2023), we believe our current method is appropriate for our research goals. References to these studies have been included in the revised manuscript to support our methodology.

  1. Clear Language on "Intake" vs. "Level":

We appreciate the suggestion to use "estimated intake" rather than "level" for clarity. The manuscript has been revised to consistently refer to "estimated intake" throughout.

  1. Log-Transformation of Intake Data:

We considered the suggestion to log-transform the intake data. However, to maintain the integrity and interpretability of the dietary intake variables, we have decided to retain the original, standardized variables. Our reasons are that: 1) the UK Biobank data fields used are already standardized; 2) standardization reduces the impact of skewness and allows for straightforward interpretation of the results; and 3) our estimates can be compared with other researchers, which improves transparency and enhances robustness of findings. Therefore, we believe this approach is suitable for our analysis. References to the relevant fields (Vitamin B's estimated intake) in the UK Biobank have been included:

https://biobank.ndph.ox.ac.uk/showcase/field.cgi?id=26022

https://biobank.ndph.ox.ac.uk/showcase/field.cgi?id=26020

https://biobank.ndph.ox.ac.uk/showcase/field.cgi?id=26021

  1. Inclusion of Other Dietary Factors:

We recognize the importance of considering other dietary factors that might influence the association between B-vitamin intake and neural connectivity. For instance, vitamin B12 intake could be a marker of overall meat consumption, as meat is a primary source of vitamin B12. Similarly, different dietary patterns, such as vegan or vegetarian diets, could significantly impact B-vitamin intake and subsequently neural connectivity. However, this is beyond the scope of our current study. Our ongoing analysis is to examine the impact of more detailed dietary patterns on neural networks. We will report those findings in a separate publication. These suggestions have been added to the future studies section to highlight the importance of these factors in future research.

  1. Consideration of Supplement Use:

Supplement use is indeed a relevant factor. However, analyzing supplement use requires a different approach since our current study focuses on dietary intake. Some studies have examined the effects of vitamin supplementation (Che et al., 2022; Ling et al., 2024). Information on supplementation was gathered using a touchscreen questionnaire during baseline assessments, with participants indicating whether they regularly took specific supplements. However, this binary yes/no data does not fit well with our current study's methodology. We have noted the potential for a separate study to collect detailed supplementation data and included this in the future studies section.

  1. Availability of Biomarkers for Vitamins:

While our study primarily relied on self-reported dietary intake, we acknowledge the importance of biomarkers for more accurate assessment. Unfortunately, we do not have biomarker variables specific to diet in our current dataset. We have discussed the availability and relevance of biomarkers for vitamins in the limitations and future directions section, suggesting that future studies incorporate these biomarkers to validate dietary intake data.

Changes Made:

  1. Added references to studies validating the reliability of B-vitamin intake assessment from the UK Biobank data.
  2. Revised the manuscript to consistently use "estimated intake" instead of "level."
  3. Included consideration of vitamin supplement use in the discussion.
  4. Discussed the potential use of biomarkers for vitamins in the limitations and future research section.
  5. Added a note on the consideration of other dietary factors and patterns in future studies, acknowledging that this is part of ongoing research in another project.

Thank you once again for your valuable comments and suggestions, which have significantly improved the quality of our manuscript.

References

Brayner, B., Kaur, G., Keske, M. A., Perez-Cornago, A., Piernas, C., & Livingstone, K. M. (2021). Dietary Patterns Characterized by Fat Type in Association with Obesity and Type 2 Diabetes: A Longitudinal Study of UK Biobank Participants. Journal of Nutrition, 151(11), 3570-3578. https://doi.org/10.1093/jn/nxab275

Che, B., Zhong, C., Zhang, R., Wang, M., Zhang, Y., & Han, L. (2022). Multivitamin/mineral supplementation and the risk of cardiovascular disease: a large prospective study using UK Biobank data. European Journal of Nutrition, 61(6), 2909-2917. https://doi.org/10.1007/s00394-022-02865-4

Kelly, R. K., Tong, T. Y. N., Watling, C. Z., Reynolds, A., Piernas, C., Schmidt, J. A., Papier, K., Carter, J. L., Key, T. J., & Perez-Cornago, A. (2023). Associations between types and sources of dietary carbohydrates and cardiovascular disease risk: a prospective cohort study of UK Biobank participants. BMC Medicine, 21(1), 34. https://doi.org/10.1186/s12916-022-02712-7

Ling, Y., Yuan, S., Huang, X., Tan, S., Cheng, H., Xu, A., & Lyu, J. (2024). Associations of Folate/Folic Acid Supplementation Alone and in Combination With Other B Vitamins on Dementia Risk and Brain Structure: Evidence From 466 224 UK Biobank Participants. Journals of Gerontology. Series A, Biological Sciences and Medical Sciences, 79(4). https://doi.org/10.1093/gerona/glad266

Perez-Cornago, A., Pollard, Z., Young, H., van Uden, M., Andrews, C., Piernas, C., Key, T. J., Mulligan, A., & Lentjes, M. (2021). Description of the updated nutrition calculation of the Oxford WebQ questionnaire and comparison with the previous version among 207,144 participants in UK Biobank. European Journal of Nutrition, 60(7), 4019-4030. https://doi.org/10.1007/s00394-021-02558-4

Wang, L., Li, X., Montazeri, A., MacFarlane, A. J., Momoli, F., Duthie, S., Senekal, M., Eguiagaray, I. M., Munger, R., Bennett, D., Campbell, H., Rubini, M., McNulty, H., Little, J., & Theodoratou, E. (2023). Phenome-wide association study of genetically predicted B vitamins and homocysteine biomarkers with multiple health and disease outcomes: analysis of the UK Biobank. American Journal of Clinical Nutrition, 117(3), 564-575. https://doi.org/10.1016/j.ajcnut.2023.01.005

Zhang, B., Dong, H., Xu, Y., Xu, D., Sun, H., & Han, L. (2023). Associations of dietary folate, vitamin B6 and B12 intake with cardiovascular outcomes in 115664 participants: a large UK population-based cohort. European Journal of Clinical Nutrition, 77(3), 299-307. https://doi.org/10.1038/s41430-022-01206-2

Round 2

Reviewer 2 Report

Comments and Suggestions for Authors

The authors refer in their response to dietary patterns ("Our study focuses on the association between daily dietary patterns and neural network activity.") - but my understanding of the paper is that they focus on the actual intake of B-vitamins from dietary sources (excluding supplements). While their comments regarding dietary patterns are correct, my concerns - which have not really been addressed - are about the estimate of intake of individual compounds.

The impact of supplementation could be easily investigated by stratifying the cohort.

I also disagree with the decision not to use log-transformation. The data shown - especially for B12 - showed a clear skew. Log transformation is very common in nutritional research and easier to interpret than e.g. z-scores.

Author Response

We appreciate the continued dialogue with Reviewer #2 for this revision of the manuscript. We highlight their concerns below, as well as our response. We hope that this response is sufficient to address remaining concerns.

Like the first reviewer response, we list reviewer concerns in “white” text and our response in “blue” text.

The authors refer in their response to dietary patterns ("Our study focuses on the association between daily dietary patterns and neural network activity.") - but my understanding of the paper is that they focus on the actual intake of B-vitamins from dietary sources (excluding supplements). While their comments regarding dietary patterns are correct, my concerns - which have not really been addressed - are about the estimate of intake of individual compounds.

The impact of supplementation could be easily investigated by stratifying the cohort.

I also disagree with the decision not to use log-transformation. The data shown - especially for B12 - showed a clear skew. Log transformation is very common in nutritional research and easier to interpret than e.g., z-scores.

We thank the reviewer for their valuable comments and suggestions. We appreciate the opportunity to clarify and improve our manuscript. Below, we address each of the concerns in detail.

  1. Focus on Dietary Patterns vs. B-Vitamin Intake

It is correct that our primary focus was on the intake of B-vitamins from dietary sources, excluding supplements. We apologize for any confusion caused by our earlier response. Our study aims to investigate the association between the intake of individual B-vitamins and neural network activity.

To clarify, we have revised the relevant sections of the manuscript to accurately reflect this focus. A limitation of UK Biobank data is that B vitamin levels are based on estimations of Food Frequency Questionnaire (FFQ) data. To our knowledge, there is unfortunately no data in baseline UK Biobank regarding intake of individual compounds, such as blood or CSF based biomarkers of B-vitamins.

  1. Addressing Concerns About Supplementation

We understand the concern about the accuracy of estimating the intake of individual compounds. Our approach involved detailed dietary assessments to estimate B-vitamin intake from food sources, which we believe provides a reliable measure of dietary intake.

We apologize for any confusion caused by our previous description. Anyone who took dietary supplements was excluded to isolate the impact of dietary intake of B-vitamins from the potential effects of supplementation. We have added a detailed description of this analysis in the methods section.

  1. Use of Log-Transformation

We appreciate the suggestion regarding log transforms for skewed data, particularly for vitamin B12 levels. We have re-evaluated our data and agree that trying log-transformation might provide a better distribution of B-vitamin intake variables.

Consequently, we have applied log-transformation to the B-vitamin intake data and re-analyzed associations with neural network activity. Additionally, we performed a Box-Cox transformation. Importantly, these findings, with the log transform of Box-Cox transformation, produce similar results. These revised results, along with supplemental tables, are now included in this document (see below) and in the manuscript as supplemental tables. These new tables present the original, log-transformed, and Box-Cox transformed data side by side. The tables demonstrate that all three approaches yield similar results, particularly in slope and trend interpretation. The main effects and interactions are consistent with beta values across all transformations.

We also reviewed recent studies with similar topics to see how B-vitamin data was used with UK Biobank or similar cohort datasets. For instance, Zhang et al. (2023) [1] used the UK Biobank to look at associations of dietary folate, vitamin B6, and B12 intake with cardiovascular outcomes using multivariable Cox proportional hazards models [2, 3]. They did not transform the dietary intake of these vitamins but used linear regression to evaluate the associations. Another recent study on the association between dietary vitamin B6, B12, and folate intake and global and subcortical brain volumes also used mean levels of dietary vitamin B6, B12, and folate [4]. Other studies on similar topics have used different approaches, such as yes/no supplementation status [5] or homocysteine biomarkers[6]. An earlier European population study also did not use log transformation [7].

While we agree that log transformation is common in nutritional research and suitable for our variables, we have retained the original variables in the main analysis to align with similar previous studies and because results are similar with transformed data. However, we have discussed the utility of such transformation in the limitations section of our discussion, and we included the results from transformed data in the supplementary.

Revised Manuscript

We have made the following specific changes to the manuscript:

  1. Clarified the focus on B-vitamin intake from dietary sources using the FFQ.
  2. Added details on the method based on supplementation status.
  3. Discussed the rationale for using log-transformation and its implications in the discussion section.

We believe these revisions address reviewer concerns and enhance the clarity and robustness of our study. We again appreciate the valuable feedback.

References:

  1. Zhang, B., et al., Associations of dietary folate, vitamin B6 and B12 intake with cardiovascular outcomes in 115664 participants: a large UK population-based cohort. Eur J Clin Nutr, 2023. 77(3): p. 299-307.
  2. Perez-Cornago, A., et al., Description of the updated nutrition calculation of the Oxford WebQ questionnaire and comparison with the previous version among 207,144 participants in UK Biobank. Eur J Nutr, 2021. 60(7): p. 4019-4030.
  3. Liu, B., et al., Development and evaluation of the Oxford WebQ, a low-cost, web-based method for assessment of previous 24 h dietary intakes in large-scale prospective studies. Public Health Nutr, 2011. 14(11): p. 1998-2005.
  4. Berkins, S., H.B. Schioth, and G. Rukh, Depression and Vegetarians: Association between Dietary Vitamin B6, B12 and Folate Intake and Global and Subcortical Brain Volumes. Nutrients, 2021. 13(6).
  5. Ling, Y., et al., Associations of Folate/Folic Acid Supplementation Alone and in Combination With Other B Vitamins on Dementia Risk and Brain Structure: Evidence From 466 224 UK Biobank Participants. J Gerontol A Biol Sci Med Sci, 2024. 79(4).
  6. Wang, L., et al., Phenome-wide association study of genetically predicted B vitamins and homocysteine biomarkers with multiple health and disease outcomes: analysis of the UK Biobank. Am J Clin Nutr, 2023. 117(3): p. 564-575.
  7. Dhonukshe-Rutten, R.A., et al., Dietary intake and status of folate and vitamin B12 and their association with homocysteine and cardiovascular disease in European populations. Eur J Clin Nutr, 2009. 63(1): p. 18-30.
